# Machine learning in predicting respiratory failure in patients with COVID-19 pneumonia —Challenges, strengths, and opportunities in a global health emergency

Davide Ferrari[1,2], Jovana Milic[1,3], Roberto Tonelli[3,4], Francesco Ghinelli[2], Marianna Meschiari[5], Sara Volpi[5], Matteo Faltoni[4], Giacomo Franceschi[5], Vittorio Iadisernia[5], Dina Yaacoub[5], Giacomo Ciusa[5], Erica Bacca[5], Carlotta Rogati[5], Marco Tutone[5], Giulia Burastero[5], Alessandro Raimondi[5], Marianna Menozzi[5], Erica Franceschini[5], Gianluca Cuomo[5], Luca Corradi[5], Gabriella Orlando[5], Antonella Santoro[5], Margherita Digaetano[5], Cinzia Puzzolante[5], Federica Carli[5], Vanni Borghi[5], Andrea Bedini[5], Riccardo Fantini[4], Luca Tabbì[4], Ivana Castaniere[3,4], Stefano Busani[6], Enrico Clini[4,7], Massimo Girardis[1,6], Mario Sarti[8], Andrea Cossarizza[7], Cristina Mussini[1,5], Federica Mandreoli[2], Paolo Missier[9‡], Giovanni Guaraldi[1,5‡]*

1 Department of Surgical, Medical, Dental and Morphological Sciences, University of Modena and Reggio Emilia, Modena, Italy, 2 Department of Physical, Computer and Mathematical Sciences, University of Modena and Reggio Emilia, Modena, Italy, 3 Clinical and Experimental Medicine PhD Program, University of Modena and Reggio Emilia, Modena, Italy, 4 Respiratory Diseases Unit, Azienda Ospedaliero-Universitaria Policlinico of Modena, Modena, Italy, 5 Department of Infectious Diseases, Azienda Ospedaliero-Universitaria Policlinico of Modena, Modena, Italy, 6 Department of Anesthesia and Intensive Care Unit, Azienda Ospedaliero-Universitaria Policlinico of Modena, Modena, Italy, 7 Department of Medical and Surgical Sciences for Children and Adults, University of Modena and Reggio Emilia, Modena, Italy, 8 Clinical Microbiology, Ospedale Civile di Baggiovara, Modena, Italy, 9 School of Computing, Newcastle University, Newcastle upon Tyne, United kingdom

‡ These authors share senior authorship.
* giovanni.guaraldi@unimore.it

**Data Availability Statement:** All relevant data are within the manuscript and its Supporting Information files.

## Abstract

### Aims

The aim of this study was to estimate a 48 hour prediction of moderate to severe respiratory failure, requiring mechanical ventilation, in hospitalized patients with COVID-19 pneumonia.

### Methods

This was an observational prospective study that comprised consecutive patients with COVID-19 pneumonia admitted to hospital from 21 February to 6 April 2020. The patients' medical history, demographic, epidemiologic and clinical data were collected in an electronic patient chart. The dataset was used to train predictive models using an established machine learning framework leveraging a hybrid approach where clinical expertise is applied along-side a data-driven analysis. The study outcome was the onset of moderate to severe respiratory failure defined as $PaO_2/FiO_2$ ratio <150 mmHg in at least one of two consecutive arterial blood gas analyses in the following 48 hours. Shapley Additive exPlanations values

**Funding:** The author(s) received no specific funding for this work.

**Competing interests:** The authors have declared that no competing interests exist.

were used to quantify the positive or negative impact of each variable included in each model on the predicted outcome.

## Results

A total of 198 patients contributed to generate 1068 usable observations which allowed to build 3 predictive models based respectively on 31-variables signs and symptoms, 39-variables laboratory biomarkers and 91-variables as a composition of the two. A fourth "boosted mixed model" included 20 variables was selected from the model 3, achieved the best predictive performance (AUC = 0.84) without worsening the FN rate. Its clinical performance was applied in a narrative case report as an example.

## Conclusion

This study developed a machine model with 84% prediction accuracy, which is able to assist clinicians in decision making process and contribute to develop new analytics to improve care at high technology readiness levels.

## Background

COVID-19 pandemic found all health care system inadequately prepared and urges the need for new tools to face this unprecedented public health and clinical emergency. The clinical complexity of COVID-19 ranges from asymptomatic cases to severe pneumonia [1–3] whose progression to respiratory failure is difficult to predict. Pneumonia mostly occurs in the second or third week of a symptomatic infection and it is characterized by a mortality rate of 3–10%, which increases risk of multiorgan failure and mechanical ventilation [4]. Patients most commonly report the sudden onset of dyspnea during daily activities or rest. Prominent clinical signs include respiratory rate $\geq$ 30 breaths per minute, blood oxygen saturation $\leq$ 93%, partial pressure of arterial oxygen to fraction of inspired oxygen ratio ($PaO_2/FiO_2$) $<$ 300 mmHg. This is an initial phase of acute respiratory distress syndrome (ARDS) that progressively leads to moderate to severe respiratory failure [4].

Overall, there is a high degree of uncertainty both in the progression of the patient's health status and in the speed at which patients develop respiratory failure requiring mechanical ventilation.

Machine learning methods such as those employed to create the model have shown potential to produce predictive models that can be applied to assist and improve clinical decisions for a broad variety of outcomes [5, 6], and have recently been used in response to the COVID-19 emergency [7–9].

However, the most comprehensive review to date [9] finds that some risk prediction models attempt to predict risk of intensive care unit admission, ventilation, intubation, but most of these studies have shortcomings (high bias, poor reporting) that make them unsuitable for clinical decision-making. In contrast, the models presented in this work are *explainable*, meaning that they provide an easily understandable grounding for the choice of predictors and their relative importance on *individual* outcomes.

The aim of this study was to have a 48-hour prediction of moderate to severe respiratory failure, requiring mechanical ventilation, in hospitalized patients with COVID-19 pneumonia.

## Methods

### Study design

This observational prospective single center study included consecutive adult patients ($\geq$18 years) admitted to Infectious Disease Clinic of the University Hospital of Modena, Italy from 21 February to 6 April 2020 with radiologically findings suggestive for COVID-19 pneumonia and confirmed by PCR method on nasopharyngeal swab.

All patients received treatment according to the Italian Society of Infectious Diseases' Guidelines (SIMIT) recommendations [10] including oxygen supply to target SaO2 > 90%; hydroxychloroquine with or without azithromycin, and low molecular weight heparin. Lopinavir/ritonavir or darunavir/cobicistat was also used up to 18 March, when a clinical trial on the former did not show any benefit of protease inhibitors against the standard of care [11].

The study outcome was the onset of moderate to severe respiratory failure defined as $PaO_2/FiO_2$ ratio < 150 mmHg ($\leq$ 13.3 kPa) in at least one of two consecutive arterial blood gas analyses in the following 48 hours.

### Data source

Hospitalized patients with COVID-19 pneumonia were included if they had at least two arterial blood gas analyses measurements in the following 48 hours.

The patients' full medical history including chronic comorbidities, demographic and epidemiological data were obtained at the hospital admission. Clinical data with signs and symptoms and complete blood count, coagulation, inflammatory and biochemical markers were routinely collected in the electronic patient charts.

Given the strict time dependent outcome definition, out of the initial sample of 295 patients and 2,889 data points available, 198 patients contributed to generate 1068 valuable observations. In detail, 603 observations contributed to the definition of respiratory failure ($PaO_2/FiO_2$ < 150 mmHg) and 465 did not meet this definition.

In the data collection period, the dataset was growing daily with the average of 84 new records per day, with a mean of 10 new data points/patient. Each data point included a complex record of observations from multiple categories: (1) signs and symptoms, (2) blood biomarkers, (3) respiratory assessment with $PaO_2/FiO_2$, (4) history of comorbidities (available in a subset of 119 patients). Some variables were collected daily, and others were recorded upon clinical indications.

All patients provided verbal, not written, informed consent due to isolation precautions. The study was approved by Regional ethical committee of Emilia Romagna (Area Vasta Nord protocol 426/2020).

### Prediction models / machine learning methods

To be considered viable for clinical use, a predictive model must not only be accurate, it must also be (1) *parsimonious*, that is, it must achieve its accuracy using the minimal number of variables; (2) robust to missing data, an important feature in clinical emergency setting where not all observations are complete at each assessment; (3) *transparent*, in the sense that the model reveals the relative importance of each variable for each prediction it makes, which may be different for different patients. This is particularly important in clinical settings as it enables healthcare professionals to interpret the pathophysiological relationships between variables, arguably resulting in an increased trust in the model's predictions.

Finally, the model should (4) minimize the number of false negatives (FN), that is, the risk of under-estimating the severity of a patient's condition.

To address the first requirement, the study produced a suite of four competing candidate models, based on aggregations of the observations into different datasets. Specifically, Model 1 was based solely on variables for first signs and symptoms; Model 2 on blood biomarkers excluding $PaO_2/FiO_2$, and Model 3 and 4 using both sets of variables, including comorbidities. This experimental design enables a comparison of the relative predictive performance across categories of variables. Furthermore, the ranking of the variables by their predictive power makes it possible to achieve a parsimonious model by eliminating the least relevant variables, resulting in an effective yet parsimonious model.

To address the second and third requirements, the LightGBM suite of algorithms (Microsoft) [12] was used. These algorithms are based on well-known ensembles of Decision Trees, and are able to produce binary classification models (positive vs negative outcome) which tolerate missing data, and which support intelligible explanations on how the model achieves its predictions.

Meeting the final goal of minimizing FN (4) required that a specialized *loss function* be developed specifically for this task. This is a function that the algorithm must minimize in order to produce optimal predictions, and in this case it includes a tunable parameter to control the ratio of FP to FN.

For this binary classifier we used the $PaO_2/FiO_2$ ratio to derive a binary outcome for the learning task, where a positive outcome is defined as $PaO_2/FiO_2 \geq 150$, and negative $PaO_2/FiO_2 < 150$. These are referred to as the positive and negative classes, respectively.

Following best practices, the dataset was divided into two parts: the training set (75% of the data—801 samples, of which 452 with $PaO_2/FiO_2 < 150$) and a complementary test set (the remaining 25% - 267 samples, of which 151 with $PaO_2/FiO_2 < 150$), which was not used in the learning phase. This separation was stratified according to the distribution of the outcome, in order to maintain a constant ratio between positive and negative classes in each of the subsets. The training set was used as input to the ML algorithm to train the model, while the test set is used to verify the predictive performance using standard metrics. The test set provided independent *ground truth* where each instance (a patient's set of observations) was associated with one of the two possible outcomes. This test set was used to evaluate the predictive performance of the model, defined in terms of true positives (TP), true negatives (TN), false negatives (FN), false positives (FP).

Model performance was measured both on the AUROC and the sensitivity. The LightGBM algorithm not only allowed tuning of their hyper-parameters (these are the parameters that cannot be learnt by the algorithm and must be set manually) in order to maximize performance, but they also allowed more specific optimization targets than simply accuracy. For this application, clinical priority was followed to maximize the sensitivity, defined as $\frac{TP}{TP+FN}$. Standard 5- and 10-fold cross validation was used to tune the model hyperparameters to achieve this goal.

To meet requirement (3) above, the learning framework provides explanations that go beyond the simple ranking of the variables. Specifically, the framework generates SHapley Additive exPlanations (SHAP) values quantifying the impact of each variable on the predicted outcome under different perspectives and both across the entire population and for individual patients [13].

## Results

Out of a total of 295 patients, 198 with COVID-19 pneumonia were included. Clinical and laboratory characteristics are shown in Table 1. The vast majority of patients were males (69.2%)

**Table 1. Clinical and laboratory characteristics of the study population at the time of hospital admission.**

| Variable | Mean or median or % |
|---|---|
| Population (%) | 198 (100%) |
| Age, years, median (IRQ) | 62.0 (54.0–73.0) |
| Sex, males (%) | 151 (76.26%) |
| **Signs and symptoms** | |
| Cough (%) | 125 (63.33%) |
| Temperature, ˚C, median (IRQ) | 37.0 (36.25–38.0) |
| Dyspnea (%) | 135 (68.85%) |
| Diarrhea (%) | 34 (17.19%) |
| Respiratory rate per minute, median (IRQ) | 22.0 (18.75–28.0) |
| **Biomarkers** | |
| Creatinine, mg/dl, median (IRQ) | 0.92 (0.8–1.27) |
| D-Dimer, ng/ml, median (IRQ) | 1120.0 (645.0–1767.5) |
| Hemoglobin, g/dl, median (IRQ) | 13.85 (12.7–14.5) |
| Platelets, $x10^9$ per L, median (IRQ) | 181.0 (154.5–258.5) |
| White Blood Cells, $x10^{12}$ per L, median (IRQ) | 6.92 (4.92–8.47) |
| $PaO_2$, %, median (IRQ) | 65.2 (56.9–76.6) |
| $PaO_2/FiO_2$, mmHg, median (IRQ) | 256.0 (149.75–301.5) |
| Oxygen saturation, %, median (IRQ) | 94.0 (91.65–95.75) |
| Creatinine kinase, U/L, median (IRQ) | 107.0 (60.25–261.75) |
| Lactate dehydrogenase, U/L, median (IRQ) | 562.0 (458.0–663.0) |
| C-reactive protein, mg/dl, median (IRQ) | 7.35 (4.22–17.65) |

**Abbreviations:** IQR–interquartile range; $FiO_2$ –fraction of inspired oxygen; $PaO_2$ –partial arterial pressure of oxygen.

with a median age of 65 years. All patients showed relevant respiratory impairment as expressed by a median $PaO_2/FiO_2$: 262 mmHg (IQR: 150.0–316.0).

To recollect, all models are binary (yes/no) classifiers of risk of developing respiratory failure, measured using a quantitative criterion ($PaO_2/FiO_2 < 150$ mmHg) and assessed using AUC and sensitivity. Also, a specific loss function was developed to privilege models that minimize the number of false negatives (FN).

Table 2 describes the four models that were used to train and test the ML tool with the following features:

Model 1: "signs and symptoms" that included 31 variables

Model 2: "laboratory biomarkers" that included 39 variables.

Model 3: "extended mixed model" that included 91 variables.

Model 4: "boosted mixed model" that includes 20 variables.

The latter was obtained from Model 3, achieving the same performance, and equal number of FN, with only 20 variables out of 91 of Model 3. To achieve this result, the variables of Model 3 were ranked according to their mean SHAP values and a backward variable selection approach was adopted, by which each variable was excluded in turn, starting from the lowest-ranked variables, and the loss in predictive performance was measured each time. This procedure identified Model 4 as the one that minimized the number of variables (20 out of the

**Table 2. Describes the four models that were used to train and test the ML tool.**

| Table legend | | | AUROC (%) |
|---|---|---|---|
| | Actual Good Outcome | Actual Good Outcome | Training model: % |
| Actual Good Outcome | **TN** | **FP** | Test model: % |
| Actual Bad Outcome | **FN** | **TP** | |
| Model 1 –Only signs and symptoms, 31 variables | | | Model 1 |
| | Actual Good Outcome | Actual Good Outcome | Training model: 89% |
| Actual Good Outcome | 74—TN | 42 FP | Test model: 69% |
| Actual Bad Outcome | 56 FN | 95 TP | |
| Model 2 –Laboratory biomarkers, 39 variables | | | Model 2 |
| | Predicted Good Outcome | Predicted Bad Outcome | Training model: 97% |
| Actual Good Outcome | 89 | 27 | Test model: 83% |
| Actual Bad Outcome | 39 | 112 | |
| Model 3 –Extended mixed model, 91 variables | | | Model 3 |
| | Predicted Good Outcome | Predicted Bad Outcome | Training model: 99% |
| Actual Good Outcome | 95 | 21 | Test model: 85% |
| Actual Bad Outcome | 40 | 111 | |
| Model 4 –Boosted mixed model, 20 variables | | | Model 4 |
| | Predicted Good Outcome | Predicted Bad Outcome | Training model: 98% |
| Actual Good Outcome | 87 | 29 | Test model: 84% |
| Actual Bad Outcome | 40 | 111 | |

original 91) without worsening the FN rate, while achieving the best performance (AUROC = 0.84). S1 Fig specifies the description of the proportion of available data for each of the 20 variables.

For each model, Table 2 reports the Area Under the Curve (AUC) for both training and test sets. We indicate the number of TP, TN, FN, FP. Variables included in each model are listed in S1 File. Figs 1 and 2 show the top 20 ranking variables used to train Model 4. X axes show the average impact of model output magnitude, expressed by SHAP values.

## Case presentation to support customization of ML into clinical practice

In addition to the theoretical performance, anecdotal validation was also provided in a real-life case setting. We applied our model to the clinical course of a 55-year-old male who was admitted for high fever and shortness of breath due to COVID-19 pneumonia. Antiviral therapy with darunavir/cobicistat was started in addition to hydroxychloroquine. He was discharged the following day, in the absence of respiratory failure as assessed by $PaO_2/FiO_2 = 420$ mmHg. Four days later, he was readmitted to hospital with high fever (39°C), diarrhea and onset of mild respiratory failure ($PaO_2/FiO_2 = 230$ mmHg). Inflammatory biomarkers were high (CRP 18 mg/dl) with elevated neutrophils. In the following 24 hours, the patient experienced a clinically unpredictable dramatic worsening of his clinical condition due to the onset of severe respiratory distress despite adequate oxygen supply ($PaO_2/FiO_2 = 88$ mmHg, respiratory rate higher than 35 breaths per minute). He was then transferred to the Intensive Care Unit (ICU) where non-invasive mechanical ventilation with helmet in pressure support mode was initiated. After 8 days of assisted spontaneous breathing, he was weaned from NIV and discharged the following day without oxygen supply (Fig 3).

As shown in the case scenario presented above, the model was retrospectively applied in order to explore the prediction of "respiratory crush" in support of clinical judgment. From the physician's point of view, the first discharge was motivated by the stable clinical conditions.

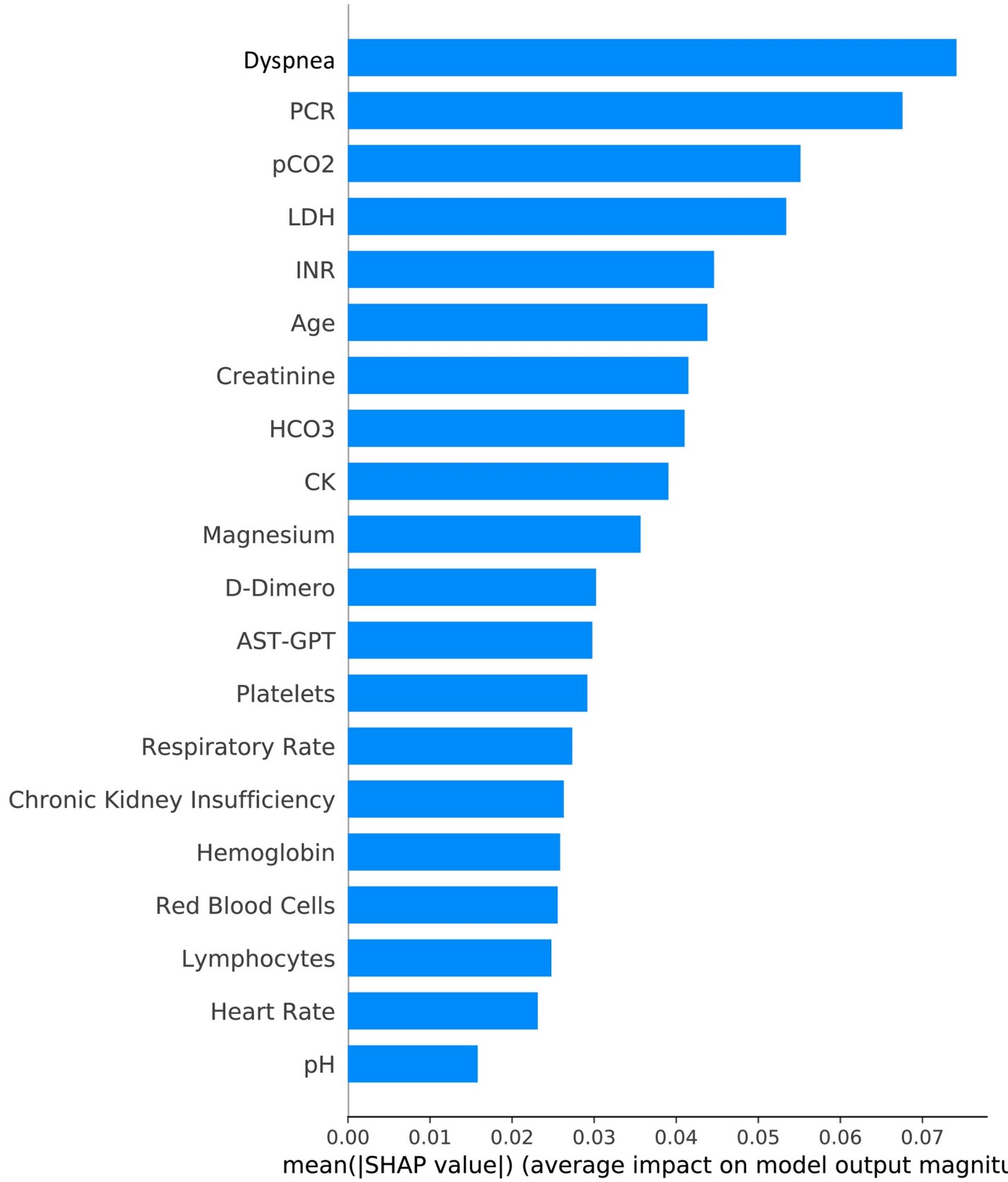

**Fig 1. Shows top 20 ranking variables used to train Model 4.** X axes show the average impact of model output magnitude, expressed by SHAP values.

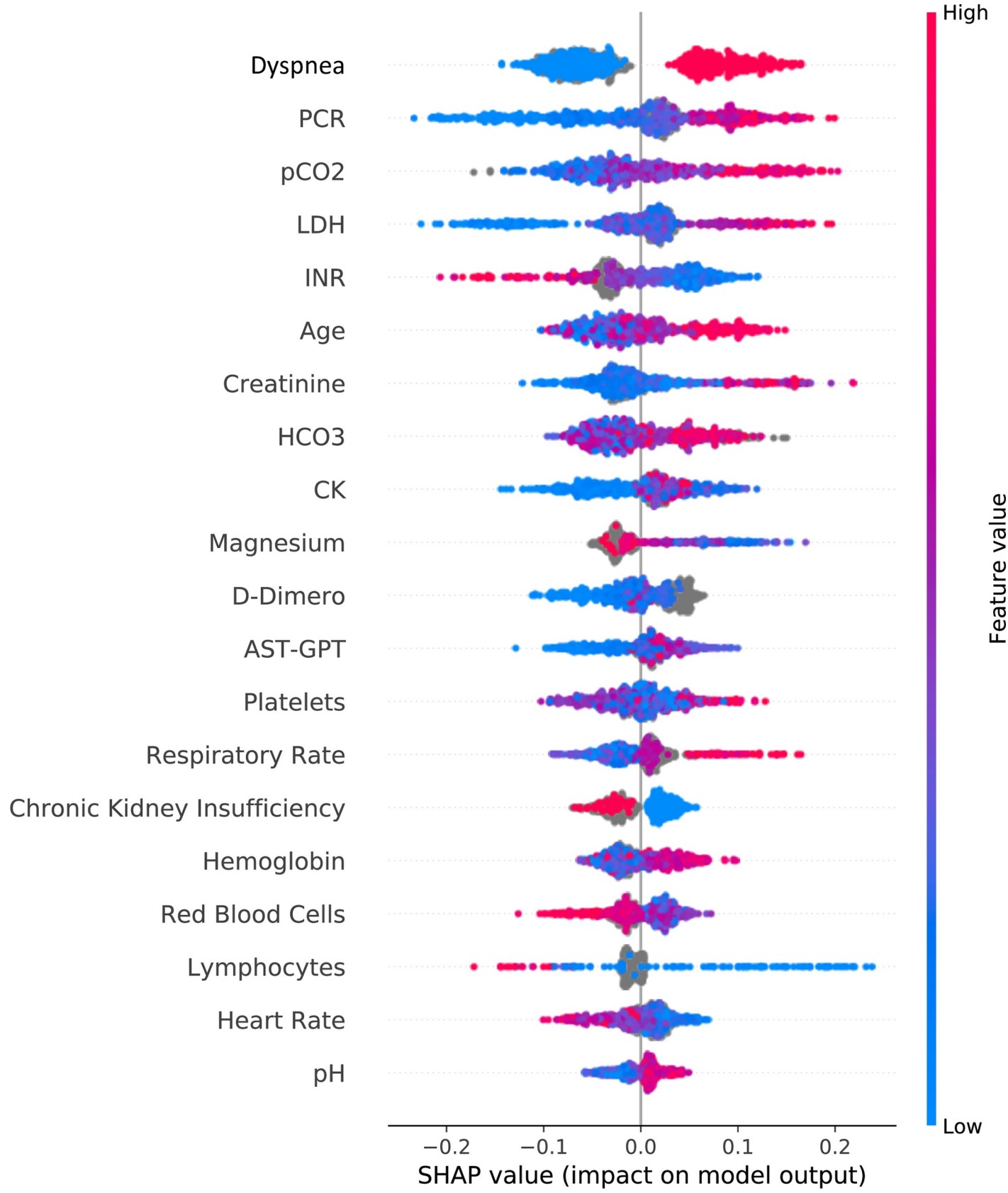

**Fig 2. The individual SHAP values for the 20 top variables.** Values of each variable may have a positive or negative impact depending on their SHAP value, for instance high values of dyspnea in red contribute strongly to the positive class (negative patient outcome), while low values in blue contribute strongly to the negative class (positive outcome).

However, our model showed a 36.6% probability of worsening of the respiratory function in the following 48 hours, meeting the criteria for mechanical ventilation with pressure support in the next two days. Moreover, the model was able to predict at the time of the second admission, the respiratory function decline that our patient actually experienced 24 hours later. The model at day 14 predicted a 47.4% risk of new worsening, but this should have been integrated with clinical data suggested by patient's perception of improvement and rapid increase of blood gas exchange. A development of our support model at the time into our clinical practice would have provided support to clinical judgment, suggesting against the first discharge, and furthermore, recommending continuous monitoring once the patient was readmitted in order to possibly avoid ICU with urgent treatment.

## Discussion

We have created a statistical learning model to assist clinicians in forecasting patients with COVID-19 who develop respiratory failure requiring mechanical ventilation. The model provide a reliable 48 hours prediction of moderate to severe respiratory failure, with an accuracy of 84% that minimizes the FN rate.

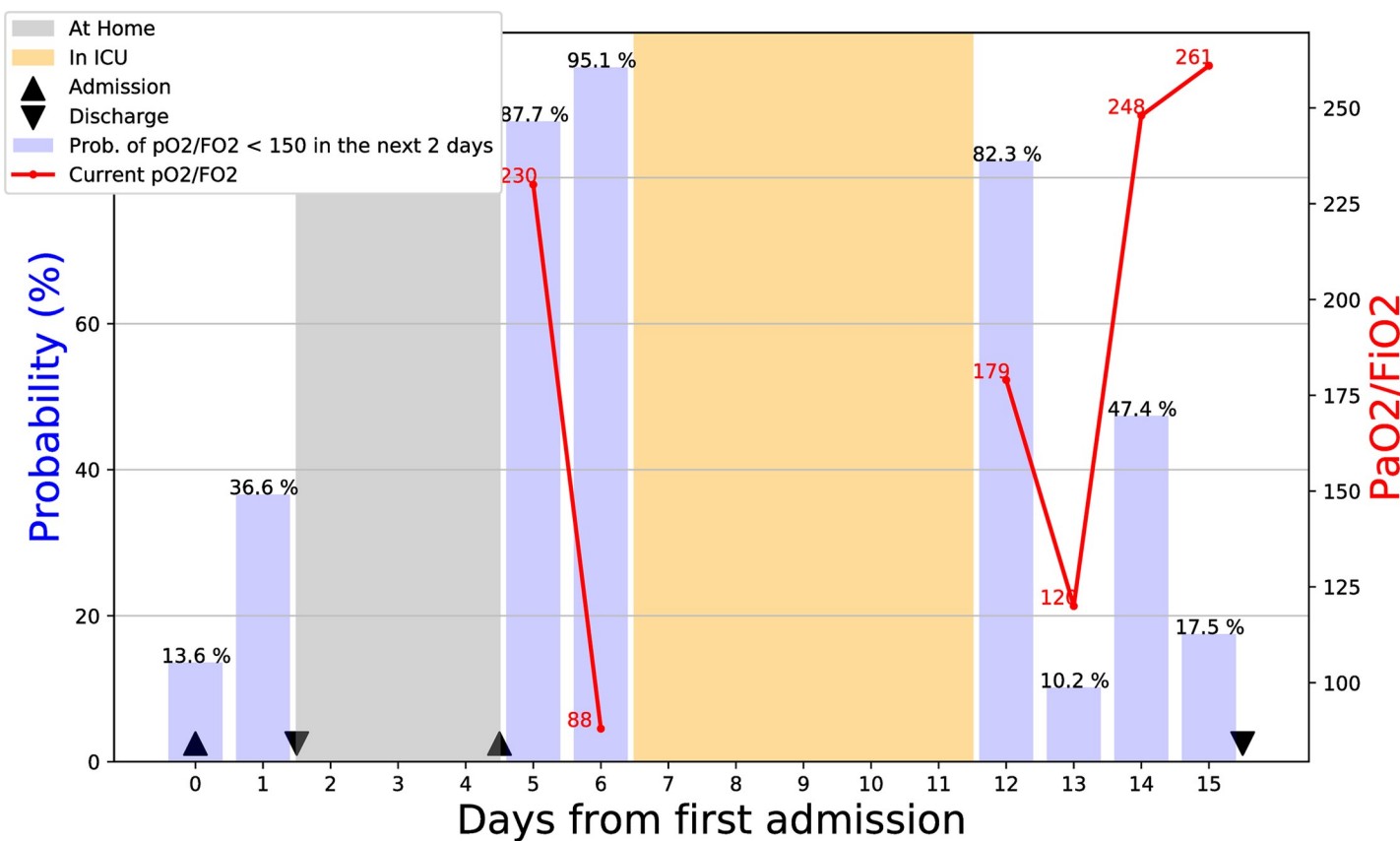

**Fig 3. Implementation of machine learning model in the case of 55-year old patient who was admitted and discharged the following day.** On day 4, the patient was re-admitted with mild respiratory insufficiency that had a 87.7% probability to experience a respiratory crush in the following 48 hours.

The level of performance of our model is in line with other ML tools used in different areas of medicine [14, 15] and it is very useful in COVID-19 clinical context where disease progression remains unpredictable both in the early virologic and in the late inflammatory phase.

It must be acknowledged that risk probability generated from the algorithm at each time point is not a measure of overall performance of the model. Clinicians should not interpret the punctual probability score as a diagnosis but rather to asses the trend measure, integrating the data in the context of clinical judgment.

We chose to have a short-term outcome to support clinicians at hospital admission and discharge. Given the rapid and dynamic clinical changes affecting COVID-19 patients, this time frame should be considered crucial for the initiation of therapies aimed at avoiding ICU admission and mechanical ventilation. In the future we may be able to develop similar models to also support clinicians to better interpret patient's clinical improvement after they are discharged from hospital.

The construction of different models followed a clinically oriented variables choice. The first model based on 31 variables obtained from signs and symptoms returned a suboptimal prediction accuracy. Adding biomarkers including respiratory variables significantly increased the forecasting capacity of the model. The best performance was obtained in the boosted mixed model, which however still requires about 20 variables. From a physician's perspective, a cluster of 20 variables may be difficult to manage in routine clinical practice. What our approach offers in support to the decision-making process is a simple interpretation of the predictions.

Machine learning approach is at the top of the list of the research priorities proposed by the Horizon 2020 program (H2020) [16]. This study may contribute to develop new analytics to improve care at high technology readiness levels.

Moderate to severe respiratory failure was chosen as an outcome being the most relevant time point in the natural history of severe COVID-19 pneumonia. At a clinical level, it represents the so-called "respiratory crush" which marks acute lung injury and leads to mechanical ventilation in ICU. At a public health level, this machine learning model might be helpful in optimizing scarce resources like ventilators and ICU beds.

A few clinical risk scores have been developed and validated to predict the occurrence of critical illness in hospitalized patients with COVID-19. These scores used at time of admission either the neutrophil/lymphocyte ratio or 10 clinical variables including radiological findings to predict critical illness using a traditional statistical approach to generate a prediction algorithm [17, 18].

In a similar experience from China, ML was used to predict mortality in patients with COVID-19, using three biomarkers only [7].

Recent data suggests that COVID-19 does not affect only respiratory system, but also other organs, such as liver, kidneys, gut, heart and central nervous system [19].

Given the multisystemic nature of this disease, limited number of parameters may not be sufficient to predict worsening in these patients.

Not surprisingly, this hard endpoint can be predicted with a very limited number of biomarkers, reducing the clinical parameters to be monitored. However, clinical worsening seems to be more challenging to forecast. An intermediate dynamic event with multiple biomarkers appears to be more difficult to predict than a final static event, such as mortality, with a small number of variables.

This science data faced several methodological challenges. Features which fed the model were chosen based both on the Shapley Values approach and on clinicians' suggestion, in a hybrid approach. This allowed to take advantage of both aspects: on one side the clinical

experience of physicians who selected variables and outcomes using a knowledge-based approach, and on the other side, the probabilistic nature of a data-driven framework.

Microsoft LightGBM framework was chosen in particular to support missing data deriving from a clinical setting where it was not practical to collect all observations at each data point. Clinicians appreciated the "Glass box" opportunity, which showed the top variables, trusting a model in which pathophysiological interpretation could still be plausible. With regards to the 20 variables selected in the hybrid models, some can be clustered within the hypoxic damage (dyspnea, $HCO3^-$, pH, reparatory and heart rate) other in relation to inflammation (C-reactive protein, D-dimer, platelets, red blood cells, lymphocytes), other in relation to organ damage (lactate dehydrogenase, creatinine-kinase). Medical Decision Support Systems must provide transparency to explain how the predictive model behaves. In this perspective, an interpretation approach is necessary, both to have better understanding of the patient's health status and to better identify dangerous biases that the model could have learnt.

The approach in our study is substantially different from traditional models based on logistic regression [20]. Risk scores derived from logistic regression mixed effect models are knowledge-driven approaches where a score is assigned by an expert to each of the limited number of selected variables [21]. In contrast, our predictive model is data driven. It is based on Decision Trees, in which the relative effect of all available variables is considered with their raw value instead of an arbitrary summarization.

This approach is particularly appropriate in COVID-19 context as it aligns well with principles of personalized medicine. It allows to identify what are the most relevant biomarkers for each individual patient which in a single patient can lead to a favorable or un-favorable progression, as shown by the Shapley Values application in Fig 3. This confirms the expected relative importance (ranking) of the variables used as features and the roles of those features in the model. This is a further proof that the model is robust as it relies on the values of variables deemed relevant in this context without actually knowing their semantics.

Our study has several limitations. Firstly, it does not take account of radiological findings on X-rays or CT, as these data were not collected consistently during hospital stay. Nevertheless, a recent radiological study demonstrated only a slight increase of prediction of respiratory failure adding thoracic CT to clinical data [22].

Secondly, we chose as outcome a cut-off of $PaO_2/FiO_2$ which represents one of the most important criteria for invasive mechanical ventilation, but not mechanical ventilation itself. Also, the model is oblivious to a whole panel of interleukins values, as they were not collected on daily basis regardless of their potential involvement in the development of severe respiratory failure. Lastly, the model will need to evolve with the growth of the dataset, providing a more accurate cut-off risk value.

In conclusion, this study developed a machine learning algorithm aimed to assist clinicians in dealing with COVID-19 health emergency. It is proving useful in predicting severe respiratory failure requiring mechanical ventilation in the following 48 hours, allowing to anticipate urgent events potentially improving management of critically ill patients.

## Supporting information

**S1 Fig.**
(PDF)

**S1 File.**
(DOCX)

## Acknowledgments

We would like to thank to Office of clinical protocols and data management in Modena: Barbara Beghetto, Giulia Nardini, Enrica Roncaglia; Office of information and communication technologies of Policlinico di Modena: Rossella Fogliani, Grazia Righini, Mario Lugli.

## Author Contributions

**Conceptualization:** Davide Ferrari, Jovana Milic, Roberto Tonelli, Gabriella Orlando, Andrea Cossarizza, Cristina Mussini, Federica Mandreoli, Paolo Missier, Giovanni Guaraldi.

**Data curation:** Davide Ferrari, Roberto Tonelli, Sara Volpi, Matteo Faltoni, Giacomo Franceschi, Vittorio Iadisernia, Dina Yaacoub, Giacomo Ciusa, Erica Bacca, Carlotta Rogati, Marco Tutone, Giulia Burastero, Alessandro Raimondi, Marianna Menozzi, Erica Franceschini, Gianluca Cuomo, Luca Corradi, Antonella Santoro, Margherita Digaetano, Cinzia Puzzolante, Federica Carli, Vanni Borghi, Andrea Bedini, Riccardo Fantini, Luca Tabbì, Ivana Castaniere, Stefano Busani, Federica Mandreoli, Paolo Missier.

**Formal analysis:** Davide Ferrari, Francesco Ghinelli, Federica Mandreoli, Paolo Missier.

**Investigation:** Davide Ferrari, Jovana Milic, Roberto Tonelli, Francesco Ghinelli, Marianna Meschiari, Sara Volpi, Matteo Faltoni, Giacomo Franceschi, Vittorio Iadisernia, Dina Yaacoub, Giacomo Ciusa, Erica Bacca, Carlotta Rogati, Marco Tutone, Giulia Burastero, Alessandro Raimondi, Marianna Menozzi, Erica Franceschini, Gianluca Cuomo, Luca Corradi, Gabriella Orlando, Antonella Santoro, Margherita Digaetano, Cinzia Puzzolante, Federica Carli, Vanni Borghi, Andrea Bedini, Riccardo Fantini, Luca Tabbì, Ivana Castaniere, Stefano Busani, Enrico Clini, Massimo Girardis, Mario Sarti, Andrea Cossarizza, Federica Mandreoli, Paolo Missier, Giovanni Guaraldi.

**Methodology:** Davide Ferrari, Jovana Milic, Francesco Ghinelli, Federica Mandreoli, Paolo Missier, Giovanni Guaraldi.

**Project administration:** Giovanni Guaraldi.

**Resources:** Davide Ferrari, Enrico Clini, Massimo Girardis, Mario Sarti, Cristina Mussini, Paolo Missier.

**Software:** Davide Ferrari, Francesco Ghinelli, Federica Mandreoli, Paolo Missier.

**Supervision:** Davide Ferrari, Jovana Milic, Andrea Cossarizza, Cristina Mussini, Federica Mandreoli, Paolo Missier, Giovanni Guaraldi.

**Validation:** Davide Ferrari, Andrea Cossarizza, Cristina Mussini, Federica Mandreoli, Paolo Missier.

**Visualization:** Davide Ferrari, Cristina Mussini, Paolo Missier.

**Writing – original draft:** Davide Ferrari, Jovana Milic, Roberto Tonelli, Marianna Meschiari, Federica Mandreoli, Paolo Missier, Giovanni Guaraldi.

**Writing – review & editing:** Davide Ferrari, Jovana Milic, Cristina Mussini, Federica Mandreoli, Paolo Missier, Giovanni Guaraldi.

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
