## [Decision Letter · Decision Letter 0]

3 Aug 2020

PONE-D-20-16454

Machine learning in predicting respiratory failure in patients with COVID-19 pneumonia - challenges, strengths, and opportunities in a global health emergency

PLOS ONE

Dear Dr. Guaraldi,

Thank you for submitting your manuscript to PLOS ONE. After careful consideration, we feel that it has merit but does not fully meet PLOS ONE’s publication criteria as it currently stands. Therefore, we invite you to submit a revised version of the manuscript that addresses the points raised during the review process.

I apologise for the long review process due, as you will understand, to the historical moment and its compelling commitments.

Based on the comments from the reviewers and my personal revision I suggest minor revisions to be made.

We look forward to receiving your revised manuscript.

Kind regards,

Paola Faverio

Academic Editor

PLOS ONE

Additional Editor Comments:

Can the authors provide more information on the validation of the machine learning method used in this study?

2. Please clarify in your Methods section whether your study was prospective or retrospective, and whether any intervention was applied. Please also clarify in your Methods section the name of the ethics committee and approval number. Please also provide details of participant consent, and whether this was written or verbal.

3.Please clarify in your Data availability statement how other researchers may obtain the data used in the study. Please also clarify whether the model code has been or will be made available to other researchers.

4. *Please explain the rationale for the development of your model in light of recent research in this area, clearly indicating which problem with existing models you are addressing.

*Please clearly report at the beginning of your methods or results section which were the key performance measures used to establish the validity and utility of your model. Please also report clearly which statistical analysis was used to establish robustness of performance measures.

*Please note that PLOS ONE requires that experiments, statistics, and other analyses must be performed to a high technical standard and described in sufficient detail to allow for reproducibility of the study (http://journals.plos.org/plosone/s/criteria-for-publication#loc-3). To demonstrate the performance of the model, we would expect comparisons to be drawn between existing state-of-the-art methods.

5. Please upload a copy of Supporting Information Table 1 which you refer to in your text on page 8.

Reviewers' comments:

**Comments to the Author**

1. Is the manuscript technically sound, and do the data support the conclusions?

Reviewer #1: Yes

Reviewer #2: Yes

2. Has the statistical analysis been performed appropriately and rigorously? 

Reviewer #1: Yes

Reviewer #2: Yes

3. Have the authors made all data underlying the findings in their manuscript fully available?

Reviewer #1: Yes

Reviewer #2: No

4. Is the manuscript presented in an intelligible fashion and written in standard English?

Reviewer #1: Yes

Reviewer #2: Yes

5. Review Comments to the Author

Reviewer #1: This is an interesting observational study about the use the prediction of respiratory failure in patients with COVID-19 pneumonia.

In my personal opinion, the principal limitation of the study is the lack of important data such as radiological included in the prediction that will be very important in this specific population. However, the study presents an important data and easy model to predict respiratory failure in COVID.19 patients with pneumonia. Finally, the inclusion and exclusion criteria are not totally clear in my opinion. This is an interesting and new issue.

I have minor comments:

1.- Please unify the use of the term COVID-19 throughout the manuscript

2.- Could the inclusion and exclusion criteria be better explained?

3.- Did you have the PaO2 / FiO2 data for all the patients? you had some specific protocol to have these data from all patients consecutively admitted to the emergency department or the authors included only patients who needed ICU admission

Reviewer #2: This manuscript shows the clinical utility of learning algorithms in the prognostication of COVID19 pneumonia. The Authors offer a prediction model for COVID19 patients, however I felt the main achievement of this project is to demonstrate that machine learning can be readily helpful in clinical practice for the chest physician. In the respiratory field, machine learning techniques have mainly been researched in lung cancer and chest imaging; it is probably time to explore its potential somewhere else.

There are some minor issues/suggestions:

1) In the abstract: “pao2/fio2 ratio < 150 in at least one of two consecutive ABG in the following 48h” - this should be included in the Methods section of the full text (there is something similar in the “study design” paragraph, but this is far clearer)

2) “Missing data” is mentioned several times: please provide a description of the missing data (a short summary or a graph or n. of variables with missing > 15% or 25%). It should be fine to add it as supplementary text, if allowed.

3) I was not able to find the number of patients with positive/negative outcome (paO2/FiO2 >/< 150) in both subsets: this should be provided

4) No information regarding past medical history/comorbidities: this can have an important impact on outcomes such as pao2/Fio2 status (e.g. chronic respiratory conditions). Please briefly mention this issue.

5) Discussion section, 2nd paragraph starting with “Our model is (1)...”: this is already in the methods section (remove it, or just refer to the methods)

6) Discussion, 2nd page “It might allow to optimize….”: I think I understood your point, but what is the subject? (Efforts to cut the progression to “respiratory crush” might allow to optimize…??)

7) Results section, last paragraph: “the model at day 14 predicted a 47% risk”, please make clear this was an error (something like “erroneously predicted 47%...”)

8) Results section, last paragraph “but this should have been integrated with clinical data….”: please remove this sentence (or move it to the discussion section) - results should not be commented

9) Results, last sentence starting with “A deployment of our support model….”: please just correct the english typo

10) Discussion session: looking at the baseline tables, data seem to be skewed toward less severe patients (see median values of LDH/D-Dimer/CRP/pao2-fio2 ratio); this may have an impact for the reproducibility of the results and model performance as well

11) Discussion session, sentence starting with “Not surprisingly this hard endpoint….”: I understand the Authors’ view, but not necessarily mortality is easier to predict than “softer” disease progression - It depends when we start following the patient for example, and mortality is of course “disease progression”. Please, rewrite the sentence with less emphasys; if possible try to add an alternative explanation for the higher number of variables needed compared to [7]

12) Discussion section, 2nd page: “were chosen based both on a statistical exploratory data analysis and on clinicians’ suggestion” - this should be stated in the methods section as well, adding some details about the type of “data analysis” used to select variable for the ML model

13) COVID19 outcome prediction models with fewer elements and similar diagnostic accuracy have been developed using a more “traditional” approach (LASSO, logistic…) [e.g. Wenhua Liang et al, JAMA 2020, PMID 32396163; Jingyuan Liu et al, J Transl Med 2020, PMID 32434518]. In order to give a broader perspective to the interested reader, please mention that in the discussion section along with your interpretation.

14) Figure 1 & 2: there is a typo (“dispnea”)

6. PLOS authors have the option to publish the peer review history of their article (what does this mean?). If published, this will include your full peer review and any attached files.

Reviewer #1: No

Reviewer #2: No

---

## [Author Response · Author response to Decision Letter 0]

21 Aug 2020

Modena, 14 August 2020

Dear Editor, 

We are very grateful for your constructive comments and suggestions to our paper entitled: “Machine learning in predicting respiratory failure in patients with COVID-19 pneumonia - challenges, strengths, and opportunities in a global health emergency”.

We here provide a point-by-point reply to the comments and we have incorporated the related changes in the manuscript. We thank the reviewers for their thoughtful insights which helped to significantly improve the manuscript.

PONE-D-20-16454

Machine learning in predicting respiratory failure in patients with COVID-19 pneumonia - challenges, strengths, and opportunities in a global health emergency

PLOS ONE

Dear Dr. Guaraldi,

Thank you for submitting your manuscript to PLOS ONE. After careful consideration, we feel that it has merit but does not fully meet PLOS ONE’s publication criteria as it currently stands. Therefore, we invite you to submit a revised version of the manuscript that addresses the points raised during the review process.

I apologise for the long review process due, as you will understand, to the historical moment and its compelling commitments.

Based on the comments from the reviewers and my personal revision I suggest minor revisions to be made.

If applicable, we recommend that you deposit your laboratory protocols in protocols.io to enhance the reproducibility of your results. Protocols.ioassigns your protocol its own identifier (DOI) so that it can be cited independently in the future. For instructions see: http://journals.plos.org/plosone/s/submission-guidelines#loc-laboratory-protocols

We look forward to receiving your revised manuscript.

Kind regards,

Paola Faverio

Academic Editor

PLOS ONE

Additional Editor Comments:

Can the authors provide more information on the validation of the machine learning method used in this study?

Authors’ response:

The following paragraphs refer to the validation of our model:

“Model performance was measured both on the AUROC and the sensitivity. The LightGBM algorithm not only allowed tuning of their hyper-parameters (these are the parameters that cannot be learnt by the algorithm and must be set manually) in order to maximize performance, but they also allowed more specific optimization targets than simply accuracy. For this application, clinical priority was followed to maximize the sensitivity, defined as TP/(TP+FN). Standard 5- and 10-fold cross validation was used to tune the model hyperparameters to achieve this goal.

To meet requirement (3) above, the learning framework provides explanations that go beyond the simple ranking of the variables. Specifically, the framework generates SHapley Additive exPlanations (SHAP) values quantifying the impact of each variable on the predicted outcome under different perspectives and both across the entire population and for individual patients [13].”

2. Please clarify in your Methods section whether your study was prospective or retrospective, and whether any intervention was applied. Please also clarify in your Methods section the name of the ethics committee and approval number. Please also provide details of participant consent, and whether this was written or verbal.

 Authors’ response:

According to Editor’s suggestion, this has been added in the methods.

3.Please clarify in your Data availability statement how other researchers may obtain the data used in the study. Please also clarify whether the model code has been or will be made available to other researchers.

 Authors’ response:

Data can be made available subject to a data disclosure agreement to be arranged with the Policlinico Universitario di Modena e Reggio Emilia.

4. *Please explain the rationale for the development of your model in light of recent research in this area, clearly indicating which problem with existing models you are addressing.

 Authors’ response:

We have added an explanation for the rationale to the background section. This starts by highlighting the findings from the systematic review [9].

Essentially, the review finds that some risk prediction models do exist that attempt to predict risk of intensive care unit admission, ventilation, intubation. The review concludes that most of these studies have shortcomings (high bias, poor reporting) that make them unsuitable for clinical decision-making. In contrast, the models presented in this work are explainable, meaning that they provide an easily understandable grounding for choice of predictors and their relative importance on individual outcomes.

*Please clearly report at the beginning of your methods or results section which were the key performance measures used to establish the validity and utility of your model. Please also report clearly which statistical analysis was used to establish robustness of performance measures.

 Authors’ response:

The models are binary (yes/no) classifiers of risk of developing respiratory failure, measured using a quantitative criterion (PaO2/FiO2 < 150 mmHg) and assessed using AUC and sensitivity. Also, a specific loss function was developed to privilege models that minimize the number of false negatives (FN).

This phrasing has been added at the beginning of the Results section.

*Please note that PLOS ONE requires that experiments, statistics, and other analyses must be performed to a high technical standard and described in sufficient detail to allow for reproducibility of the study (http://journals.plos.org/plosone/s/criteria-for-publication#loc-3). To demonstrate the performance of the model, we would expect comparisons to be drawn between existing state-of-the-art methods.

 Authors’ response:

Statistics, and other analyses were performed in a high technical standard described in the methods.

5. Please upload a copy of Supporting Information Table 1 which you refer to in your text on page 8.

Authors’ response:

Supplementary table 1 is now added listing in alphabetical order the 91 variables used to build the model.

Reviewers' comments:

Comments to the Author

1. Is the manuscript technically sound, and do the data support the conclusions?

Reviewer #1: Yes

Reviewer #2: Yes

2. Has the statistical analysis been performed appropriately and rigorously? 

Reviewer #1: Yes

Reviewer #2: Yes

3. Have the authors made all data underlying the findings in their manuscript fully available?

Reviewer #1: Yes

Reviewer #2: No

4. Is the manuscript presented in an intelligible fashion and written in standard English?

Reviewer #1: Yes

Reviewer #2: Yes

5. Review Comments to the Author

Reviewer #1: This is an interesting observational study about the use the prediction of respiratory failure in patients with COVID-19 pneumonia.

In my personal opinion, the principal limitation of the study is the lack of important data such as radiological included in the prediction that will be very important in this specific population. 

 Authors’ response:

We agree with this comment and in the limitation of the study it is written” Our study has several limitations. Firstly, it does not take account of radiological findings on X-rays or CT,”. Nevertheless some radiological studies underline that the contribution of radiological findings in the prediction of subsequent Acute Respiratory Distress Syndrome in minimal. In the case of Colombi study, CT findings mimimally increased AUC prediction from 0,83 to 0,85 in addition to clinical data. This sentence is added: 

“Our study has several limitations. Firstly, it does not take account of radiological findings on X-rays or CT, as these data were not collected consistently during hospital stay. Nevertheless, a recent radiological study demonstrated only a slight increase of prediction of respiratory failure adding thoracic CT to clinical data.”

However, the study presents an important data and easy model to predict respiratory failure in COVID.19 patients with pneumonia. Finally, the inclusion and exclusion criteria are not totally clear in my opinion. 

 Authors’ response:

The following sentence has been added in the methods:

“Hospitalized patients with COVID-19 pneumonia were included if they had at least two arterial blood gas analyses measurements in the following 48 hours.”

This is an interesting and new issue.

I have minor comments:

1.- Please unify the use of the term COVID-19 throughout the manuscript

 Authors’ response:

This has been corrected.

2.- Could the inclusion and exclusion criteria be better explained?

Authors’ response:

Please, see below.

3.- Did you have the PaO2 / FiO2 data for all the patients? you had some specific protocol to have these data from all patients consecutively admitted to the emergency department or the authors included only patients who needed ICU admission

Authors’ response:

Yes, this was specified in the inclusion criteria.

Reviewer #2: This manuscript shows the clinical utility of learning algorithms in the prognostication of COVID19 pneumonia. The Authors offer a prediction model for COVID19 patients, however I felt the main achievement of this project is to demonstrate that machine learning can be readily helpful in clinical practice for the chest physician. In the respiratory field, machine learning techniques have mainly been researched in lung cancer and chest imaging; it is probably time to explore its potential somewhere else.

There are some minor issues/suggestions:

1) In the abstract: “pao2/fio2 ratio < 150 in at least one of two consecutive ABG in the following 48h” - this should be included in the Methods section of the full text (there is something similar in the “study design” paragraph, but this is far clearer)

Authors’ response:

This sentence has been improved according to reviewer suggestion:

The study outcome was the onset of moderate to severe respiratory failure defined as PaO2/FiO2 ratio < 150 mmHg (≤ 13.3 kPa) in at least one of two consecutive arterial blood gas analyses in the following 48 hours.

2) “Missing data” is mentioned several times: please provide a description of the missing data (a short summary or a graph or n. of variables with missing > 15% or 25%). It should be fine to add it as supplementary text, if allowed.

Authors’ response:

This sentence and table have been added in the text:

Supplementary Figure 2 specifies the description of the proportion of available data for each of the 20 variables.

3) I was not able to find the number of patients with positive/negative outcome (paO2/FiO2 >/< 150) in both subsets: this should be provided

Authors’ response:

Study outcome was identified from PaO2/FiO2 data and individual patients could contribute to both positive and negative outcomes. The same patient could contribute to generate the definition of respiratory failure or the lack of this condition, therefor it may be misleading to specify the number of patients in non-mutually exclusive groups. We preferred to specify in the text the number of observations rather than the number of patients. It was written: “In detail, 603 observations contributed to the definition of respiratory failure (PaO2/FiO2 < 150 mmHg) and 465 did not meet this definition”. 

4) No information regarding past medical history/comorbidities: this can have an important impact on outcomes such as pao2/Fio2 status (e.g. chronic respiratory conditions). Please briefly mention this issue.

Authors’ response:

History of comorbidities was available in a subset of 119 patients and was included in the training and test set of machine learning prediction, as specified in methods.

5) Discussion section, 2nd paragraph starting with “Our model is (1) ...”: this is already in the methods section (remove it, or just refer to the methods)

Authors’ response:

This sentence has been removed.

6) Discussion, 2nd page “It might allow to optimize….”: I think I understood your point, but what is the subject? (Efforts to cut the progression to “respiratory crush” might allow to optimize…??)

Authors’ response:

This sentence has been corrected as following:

“At a public health level, this machine learning model might be helpful in optimizing scarce resources like ventilators and ICU beds.”

7) Results section, last paragraph: “the model at day 14 predicted a 47% risk”, please make clear this was an error (something like “erroneously predicted 47%...”)

Authors’ response:

47% risk is the risk probability at that time point and not a measure of performance of the model. Each time point probability, as mentioned thereafter should have been integrated with clinical data. We state “but” to stress the need to integrate prediction and clinical data. 

The following sentence was added:

“It must be acknowledged that risk probability generated from the algorithm at each time point is not a measure of overall performance of the model. Clinicians should not interpret the punctual probability score as a diagnosis but rather to assess the trend measure, integrating the data in the context of clinical judgment”.

8) Results section, last paragraph “but this should have been integrated with clinical data….”: please remove this sentence (or move it to the discussion section) - results should not be commented

Authors’ response:

This result is a case example which explain probability risk and overall performance of the model and do not fit elsewhere the discussion section. We respectfully suggest to leave this small comment in this section.

As mentioned above this sentence was added in the discussion:

“It must be acknowledged that risk probability generated from the algorithm at each time point is not a measure of overall performance of the model. Clinicians should not interpret the punctual probability score as a diagnosis but rather assessing the trend measure integrating the data in the context of clinical judgment.”

9) Results, last sentence starting with “A deployment of our support model….”: please just correct the english typo

Authors’ response:

This has been corrected.

10) Discussion session: looking at the baseline tables, data seem to be skewed toward less severe patients (see median values of LDH/D-Dimer/CRP/pao2-fio2 ratio); this may have an impact for the reproducibility of the results and model performance as well

Authors’ response:

A certain amount of skew in a multi-dimensional space of features (the variables in Table I) is inevitable. Regarding model performance, skew may affect generalization error, that is it may result in model overfitting. We control overfitting according to standard Machine Learning practices, namely by (1) deploying ensemble methods that are known to be robust to data skew, i.e., LightGBM / XGBoost in this instance, (2) using K-fold cross-validation to assess model performance, and (3) computing model performance on an independent test set that was not used for training.

11) Discussion session, sentence starting with “Not surprisingly this hard endpoint….”: I understand the Authors’ view, but not necessarily mortality is easier to predict than “softer” disease progression - It depends when we start following the patient for example, and mortality is of course “disease progression”. Please, rewrite the sentence with less emphasis; if possible try to add an alternative explanation for the higher number of variables needed compared to [7]

Authors’ response:

The sentence has been introduced as following:

“Recent data suggests that COVID-19 does not affect only respiratory system, but also other organs, such as liver, kidneys, gut, heart and central nervous system [19]. Given the multisystemic nature of this disease, limited number of parameters may not be sufficient to predict worsening in these patients. 

Not surprisingly, this hard endpoint can be predicted with a very limited number of biomarkers, reducing the clinical parameters to be monitored. However, clinical worsening seems to be more challenging to forecast. An intermediate dynamic event with multiple biomarkers appears to be more difficult to predict than a final static event, such as mortality, with a small number of variables.”

12) Discussion section, 2nd page: “were chosen based both on a statistical exploratory data analysis and on clinicians’ suggestion” - this should be stated in the methods section as well, adding some details about the type of “data analysis” used to select variable for the ML model

Authors’ response:

This sentence has been modified in the discussion.

This science data faced several methodological challenges. Features which fed the model were chosen based both on the Shapley Values approach and on clinicians' suggestion, in a hybrid approach.

13) COVID19 outcome prediction models with fewer elements and similar diagnostic accuracy have been developed using a more “traditional” approach (LASSO, logistic…) [e.g. Wenhua Liang et al, JAMA 2020, PMID 32396163; Jingyuan Liu et al, J Transl Med 2020, PMID 32434518]. In order to give a broader perspective to the interested reader, please mention that in the discussion section along with your interpretation.

Authors’ response:

We thank the reviewer for the valuable references that have been introduced in the discussion:

“A few clinical risk scores have been developed and validated to predict the occurrence of critical illness in hospitalized patients with COVID-19. These scores used at time of admission either the neutrophil/lymphocyte ratio or 10 clinical variables including radiological findings to predict critical illness using a traditional statistical approach to generate a prediction algorithm [17,18].”

14) Figure 1 & 2: there is a typo (“dispnea”)

Authors’ response:

This has been corrected.

6. PLOS authors have the option to publish the peer review history of their article (what does this mean?). If published, this will include your full peer review and any attached files.

Do you want your identity to be public for this peer review? For information about this choice, including consent withdrawal, please see our Privacy Policy.

Reviewer #1: No

Reviewer #2: No

---

## [Decision Letter · Decision Letter 1]

2 Sep 2020

Machine learning in predicting respiratory failure in patients with COVID-19 pneumonia - challenges, strengths, and opportunities in a global health emergency

PONE-D-20-16454R1

Dear Dr. Guaraldi,

We’re pleased to inform you that your manuscript has been judged scientifically suitable for publication and will be formally accepted for publication once it meets all outstanding technical requirements.

Kind regards,

Paola Faverio

Academic Editor

PLOS ONE

Additional Editor Comments:

Thank you for addressing all the issues highlighted in the revision process.

Reviewers' comments:

Reviewer's Responses to Questions

**Comments to the Author**

1. If the authors have adequately addressed your comments raised in a previous round of review and you feel that this manuscript is now acceptable for publication, you may indicate that here to bypass the “Comments to the Author” section, enter your conflict of interest statement in the “Confidential to Editor” section, and submit your "Accept" recommendation.

Reviewer #1: All comments have been addressed

Reviewer #2: All comments have been addressed

2. Is the manuscript technically sound, and do the data support the conclusions?

Reviewer #1: Yes

Reviewer #2: Yes

3. Has the statistical analysis been performed appropriately and rigorously? 

Reviewer #1: Yes

Reviewer #2: Yes

4. Have the authors made all data underlying the findings in their manuscript fully available?

Reviewer #1: Yes

Reviewer #2: Yes

5. Is the manuscript presented in an intelligible fashion and written in standard English?

Reviewer #1: Yes

Reviewer #2: Yes

6. Review Comments to the Author

Reviewer #1: The authors have responded to all my comments, and the current version of the article in my opinion can be published

Reviewer #2: I would like to thank the Authors for the changes provided. In my opinion, the revised manuscript draft has been improved and is better balanced. I don't have any further comment.

7. PLOS authors have the option to publish the peer review history of their article (what does this mean?). If published, this will include your full peer review and any attached files.

Reviewer #1: **Yes: **Catia Cilloniz

Reviewer #2: No

---

## [Editor Report · Acceptance letter]

4 Nov 2020

PONE-D-20-16454R1 

Machine learning in predicting respiratory failure in patients with COVID-19 pneumonia - challenges, strengths, and opportunities in a global health emergency 

Dear Dr. Guaraldi:

I'm pleased to inform you that your manuscript has been deemed suitable for publication in PLOS ONE. Congratulations! Your manuscript is now with our production department. 

Kind regards, 

on behalf of

Dr. Paola Faverio 

Academic Editor

PLOS ONE